# Impact of Layer Selection in Histopathology Foundation Models on Downstream Task Performance

**Witali Aswolinskiy**                              W.ASWOLINSKIY@PAICON.COM
*PAICON GmbH, Heidelberg, Germany*

**Martin Paulikat**                                ND297@UNI-HEIDELBERG.DE
*Department of Applied Tumor Biology, Universitätsklinikum Heidelberg, Heidelberg, Germany*

**Christian Aichmüller**                           C.AICHMUELLER@PAICON.COM
*PAICON GmbH, Heidelberg, Germany*

## Abstract

Self-supervised vision transformer models trained on large histopathology datasets are increasingly used as feature encoders for downstream tasks. However, their final layer might not be optimal for all tasks due to the mismatch between the pre-training and downstream objectives. We investigate the influence of the layer selection in five public, transformer-based histopathology encoders on downstream task performance both on patch- and slide-level. Our results demonstrate that choosing a different layer for feature encoding can lead to performance improvements up to eleven percent depending on the task and the model.

**Keywords:** computational pathology, self-supervised pre-training, foundation models, layer selection

## 1. Introduction

The field of computational pathology has seen a rise in self-supervised pre-training of foundational models. These models leverage vast collections of unlabeled whole slide images (WSIs) to learn a rich and generic representation of histopathological features, independent of specific downstream tasks. These models have been successfully applied to various downstream tasks across several tissue and cancer types (Lai et al., 2023).

However, a key question remains: are the learned representations from the final layer of these pre-trained encoders truly optimal for all downstream tasks? The self-supervised objective might not be perfectly aligned with the specific requirements of a particular task, potentially neglecting organ-specific features relevant for downstream prediction. This work investigates the impact of the layer selection within transformer-based foundational models on downstream task performance.

We evaluate five publicly available transformer-based histopathology encoders on three patch-level and three slide-level downstream tasks encompassing both tissue-subtyping and gene status prediction. By exploring different layers within the encoder architecture, we demonstrate that treating layer selection as a tunable hyperparameter can improve performance on downstream tasks in computational pathology.

## 2. Data and Methods

**Patch-level evaluation:** We selected three tissue classification tasks: PatchCamelyon (Veeling et al., 2018), NCT-CRC (Kather et al., 2018) and BreakHis (Spanhol et al., 2015). We additionally increased the task difficulty by reducing the number of images in Patch-Camelyon to 10,000 (half used for training) and NCT-CRC to 16,943 (8,943 used for training; also ignoring the background class). From BreakHis we used the original 1693 images at 400x magnitude (1148 used for training). All images were rescaled to 224x224px.

**Slide-level evaluation:** We used three cohorts: Camelyon16 (Bejnordi et al., 2017) with 399 slides (129 used for testing) and two cohorts from the The Cancer Genome Atlas (TCGA, Network et al. (2012)): COAD with 1216 slides to predict microsatellite instability (MSI) and BRCA with 965 slides to predict HER2 status. We trained on the TCGA slides and evaluated the MSI prediction on the CRC cohort from the Clinical Proteomic Tumor Analysis Consortium (CPTAC, Edwards et al. (2015)) with 221 slides and HER2 prediction on the Yale Pathology electronic database cohort (HER2-Yale, Farahmand et al. (2022)) with 191 slides.

**Encoders:** We evaluated five recent transformer-based histopathological vision models: CTransPath (Wang et al., 2022), Phikon (Filiot et al., 2023), two encoders from Lunit Inc. (Lunit-Dino-8 and Lunit-Dino-16, Kang et al. (2023)) and an encoder finetuned on colorectal slides from TCGA (Dino2s-CRC, Roth et al. (2024)). The ImageNet pre-trained ViT-S model with DINOv2 (Dino2s, Oquab et al. (2023)) served as our baseline. Counting the normalization layers, CTransPath has 29 layers, whereas the other encoders have 25. Due to computational restrictions, we focused on feature extraction from the final four layers for the slide-wise tasks. We used *Attention-Based Multiple Instance Learning* (ABMIL, Ilse et al. (2018)) with a dropout rate of 25% and early stopping when the validation ROC-AUC (area under the receiver operating characteristic curve) did not improve after 25 epochs. To account for potential variability, we ran ten experiments with different initializations for each layer and averaged the ROC-AUC on the test sets. For the patch-wise tasks, we evaluated the final 20 encoder layers. The encoded features were directly fed into a logistic regression classifier with the macro-F1 score as performance measure.

## 3. Results

**Patch-level tasks** As shown in Figure 1, performance on patch-wise classification tasks generally increased with deeper encoder layers. This trend was most pronounced for NCT-CRC, where peak performance was reached in one of the final four layers. Similar, though less pronounced trends were observed for the other tasks, except for BreakHis with the ImageNet pre-trained Dino2s encoder. Interestingly, the pen-ultimate layer of the Lunit-Dino-16 encoder consistently underperformed across all tasks. Otherwise, the performance variations within the final layers were modest.

**Slide-level tasks** Performance on slide-wise classification tasks varied based on the chosen encoder layer and task (Table 1). Compared to the final layer, selecting an earlier one yielded improvements of up to 1.5% for Camelyon16, 11.3% for HER2 and 1.4% for MSI.

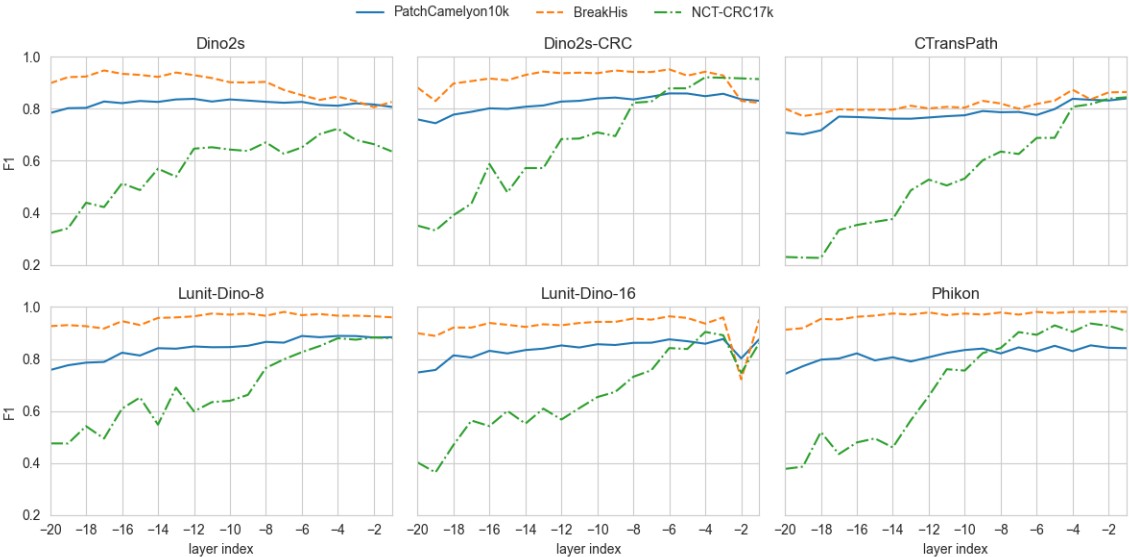

Figure 1: Patch-wise test results for the latest 20 layers ordered from early (-20) to the last (-1) with logistic regression applied directly to the layer encodings.

## 4. Discussion and Conclusion

This study investigated the impact of layer selection within transformer-based histopathology foundation models. Our findings showed marked performance differences between layers, indicating that significant gains can be achieved by selecting a task-specific layer rather than relying solely on the final one. Future work will include the evaluation of more encoders and combining encoder fine-tuning with layer selection.

| Encoder | Layer Task | -4 | -3 | -2 | -1 |
|---|---|---|---|---|---|
| CTransPath | Camelyon16 | 0.819 ± 0.037 | 0.837 ± 0.024 | 0.866 ± 0.023 | **0.907 ± 0.019** |
| Dino2s | Camelyon16 | **0.717 ± 0.032** | 0.698 ± 0.021 | 0.702 ± 0.049 | 0.706 ± 0.060 |
| Lunit-Dino-16 | Camelyon16 | 0.960 ± 0.008 | 0.963 ± 0.012 | 0.859 ± 0.018 | **0.966 ± 0.012** |
| Lunit-Dino-8 | Camelyon16 | 0.956 ± 0.028 | 0.959 ± 0.010 | **0.960 ± 0.022** | 0.948 ± 0.022 |
| Phikon | Camelyon16 | 0.956 ± 0.031 | **0.977 ± 0.007** | 0.976 ± 0.013 | 0.962 ± 0.025 |
| CTransPath | HER2-Yale | 0.711 ± 0.056 | **0.738 ± 0.038** | 0.708 ± 0.064 | 0.625 ± 0.042 |
| Dino2s | HER2-Yale | **0.580 ± 0.053** | 0.557 ± 0.020 | 0.541 ± 0.050 | 0.553 ± 0.061 |
| Lunit-Dino-16 | HER2-Yale | 0.713 ± 0.052 | **0.740 ± 0.048** | 0.737 ± 0.055 | 0.714 ± 0.039 |
| Lunit-Dino-8 | HER2-Yale | 0.726 ± 0.063 | 0.716 ± 0.053 | **0.757 ± 0.047** | 0.704 ± 0.063 |
| Phikon | HER2-Yale | 0.675 ± 0.037 | **0.699 ± 0.036** | 0.680 ± 0.038 | 0.676 ± 0.045 |
| CTransPath | MSI-CPTAC | 0.898 ± 0.029 | 0.902 ± 0.018 | 0.905 ± 0.024 | **0.921 ± 0.013** |
| Dino2-CRC | MSI-CPTAC | 0.935 ± 0.018 | 0.925 ± 0.017 | **0.940 ± 0.011** | 0.939 ± 0.011 |
| Dino2s | MSI-CPTAC | 0.761 ± 0.038 | 0.716 ± 0.040 | 0.755 ± 0.041 | **0.765 ± 0.030** |
| Lunit-Dino-16 | MSI-CPTAC | 0.891 ± 0.013 | **0.922 ± 0.016** | 0.887 ± 0.005 | 0.917 ± 0.020 |
| Lunit-Dino-8 | MSI-CPTAC | **0.916 ± 0.010** | 0.902 ± 0.017 | 0.886 ± 0.011 | 0.902 ± 0.017 |
| Phikon | MSI-CPTAC | 0.921 ± 0.016 | 0.929 ± 0.016 | 0.928 ± 0.011 | **0.933 ± 0.010** |

Table 1: Slide-wise test results for the final four layers (last one is -1), with the best layer indicated in bold. Metric: Average ROC-AUC over 10 experiments with standard deviation.

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
