# OpenReview forum: "Impact of Layer Selection in Histopathology Foundation Models on Downstream Task Performance"
_MIDL.io/2024/Short_Papers — MIDL 2024 Short Papers_

### Official Review · Reviewer_NmKR · 2024-04-16

**Confidence:** 5
**Final Rating:** 5

**Review:**

The paper demonstrates strength in its thorough investigation of layer selection within transformer-based histopathology foundation models, presenting a nuanced understanding of how different layers influence model performance. The findings reveal performance variances between layers, emphasizing the potential benefits of selecting task-specific layers over defaulting to the final layer. Additionally, the paper sets a promising direction for future research, including the exploration of more encoders and the combination of encoder fine-tuning with layer selection, further solidifying its value and relevance for publication.

---

### Decision · Program_Chairs · 2024-04-26

Accept